# Relationship between Intervertebral Disc Compression Force and Sagittal Spinopelvic Lower Limb Alignment in Elderly Women in Standing Position with Patient-Specific Whole Body Musculoskeletal Model

**DOI:** 10.3390/ijerph192416452

**Published:** 2022-12-08

**Authors:** Takanori Miura, Michio Hongo, Yuji Kasukawa, Hiroaki Kijima, Daisuke Kudo, Kimio Saito, Ryota Kimura, Takehiro Iwami, Naohisa Miyakoshi

**Affiliations:** 1Department of Orthopedic Surgery, Tazawako Hospital, Akita 014-1201, Japan; 2Department of Orthopedic Surgery, Graduate School of Medicine, Akita University, Akita 010-8543, Japan; 3Department of Rehabilitation Medicine, Akita University Hospital, Akita 010-8543, Japan; 4Department of Mechanical Engineering, Faculty of Engineering Science, Akita University, Akita 010-8502, Japan

**Keywords:** adult spinal deformity, musculoskeletal model, musculoskeletal simulation, sagittal alignment, intervertebral disc, sagittal vertical axis, T1 slope, biomechanics, postural adaptation

## Abstract

The intervertebral disc loading based on compensated standing posture in patients with adult spinal deformity remains unclear. We analyzed the relationship between sagittal alignment and disc compression force ($F_{m}$). In 14 elderly women, the alignment of the sagittal spinopelvic and lower extremities was measured. $F_{m}$ was calculated using the Anybody Modeling System. Patients were divided into low sagittal vertical axis (SVA) and high SVA groups. Comparisons between the two groups were performed and the relationship between the $F_{m}$ and each parameter was examined using Spearman’s correlation coefficient (r). The mean lumbar $F_{m}$ in the high SVA group was 67.6%; significantly higher than that in the low SVA group (*p* = 0.046). There was a negative correlation between cervical $F_{m}$ with T1 slope (r = −0.589, *p* = 0.034) and lumbar $F_{m}$ with lumbar lordosis (r = −0.566, *p* = 0.035). Lumbar $F_{m}$ was positively correlated with center of gravity-SVA (r = 0.615, *p* = 0.029), T1 slope (r = 0.613, *p* = 0.026), and SVA (r = 0.612, *p* = 0.020). The results suggested sagittal malalignment increased the load on the thoracolumbar and lower lumbar discs and was associated with cervical disc loading.

## 1. Introduction

Adult spinal deformity (ASD) is a complex disease characterized by various types of lumbar or thoracolumbar deformities [1,2]. There is growing concern surrounding this disorder as it is associated with chronic back pain, imbalance, and risk of falling and decreased health-related quality of life (QoL) [3,4,5]. Sagittal imbalance results in increased energy expenditure during the standing position, and the patients compensate to maintain an upright posture and horizontal gaze through thoracic hypokyphosis, pelvic retroversion, hip extension, knee flexion, and cervical spine extension [4,6,7,8]. Recently, cervical alignment has been focused to gain adequate horizontal gaze after corrective surgery for ASD [9]. Such regional and global sagittal malalignment has been shown to correlate with pain and disability and possibly cause deformity progression due to loss of disc height and increased load on the anterior part of the vertebral body [5,10,11,12]. However, the mechanical behavior of intervertebral disc loading in all regions of the spine according to postural changes remains unclear. Biomechanical analysis is necessary to clarify the relationship between changes in sagittal alignment and disc loading.

Common parameters used to define the spinopelvic sagittal alignment of ASD include the sagittal vertical axis (SVA), pelvic tilt (PT), and pelvic incidence (PI) minus lumbar lordosis (LL) [1,2,5,10]. In addition, due to the influence of the center of gravity (COG) on spinal balance and load distribution, biomechanical analyses should include an assessment of cervical alignment and the cranial center of mass [13,14,15,16]. Several studies have evaluated intradiscal pressure in the standing posture in the thoracolumbar or lumbar region by inserting a pressure sensor inside the intervertebral discs [17,18,19,20]. However, this procedure is invasive and only reported below the lower thoracic spine (below T6). Therefore, the musculoskeletal model (MSM) is widely used to evaluate the intervertebral disc load of the entire spinal column [21,22,23].

Among MSMs, the Anybody Modeling System (AMS) is increasingly being exploited and updated worldwide and has been validated as a full-body model for the estimation of lumbar spinal loads [24]. Furthermore, it enables detailed analysis of the cervical and thoracolumbar regions [25,26,27,28]. Taking into account the cervical spine compensation mechanism in patients with ASD and the influence of the COG due to the position of the cranial, evaluation of the intervertebral disc load using an MSM should adapt the sagittal spinal alignment, including the cervical spine.

Although there have been studies on AMS with altered thoracolumbar and pelvic alignment, there are no studies on altered cervical spine alignment [29,30,31]. Therefore, this study aimed to investigate the relationship between sagittal alignment, including cervical to lumbar spine, pelvis, and lower extremity in the standing position, and disc compression force ($F_{m}$) from the cervical to the lumbar spine analyzed by patient-specific adapted AMS that also considered cervical spine alignment in elderly women.

## 2. Materials and Methods

### 2.1. Patient-Specific Data

After institutional review board approval (approval number: 2022-4), a retrospective review was conducted to identify patients who had whole spine and lateral lower extremity radiograph. The inclusion criteria were ambulatory patients diagnosed with primary osteoporosis and no history of heavy work. Patients with a history of spine surgery, two or more vertebral fractures, hip or knee arthroplasty, or complaints of severe back pain were excluded. Finally, 14 women (78.8 ± 7.0 years) were enrolled. Written informed consent was obtained from all patients.

### 2.2. Image Acquisition and Anatomical Parameters Extraction

Lateral radiographs of the whole spine and lower extremities were obtained with both hands placed on the clavicle in a relaxed standing position. The spinopelvic lower-extremity alignment parameters were measured by a single author. The spinal parameters included in this analysis were the occipito–C2 angle (O–C2 angle; the angle between McGregor’s line and the inferior endplate line of C2) [32], C2–C7 angle (the angle between the lines parallel to the inferior endplate of C2 and C7 vertebral bodies) [33], C2–C7 SVA (the horizontal distance between the posterosuperior corner of the C7 vertebral body and a plumb line drawn from the centroid of C2) [34], COG–SVA (the distance between a plumb line dropped from the COG, drawn from the anterior portion of the external auditory canal, to the posterior superior endplate of S1) [35], T1 slope (the angle between a horizontal line and the superior endplate of T1) [36], thoracic kyphosis (TK; Cobb angle from the upper endplate of T4 to the lower endplate of T12), LL (Cobb angle from the upper endplate of L1 to the lower endplate of S1), and SVA (horizontal distance from the C7 plumb line that originates in the middle of the C7 vertebral body to the posterior superior endplate of S1). The pelvic parameters included in this analysis were PT (the angle between the line connecting the midpoint of the sacral plate to the bi-coxo-femoral axis and the vertical plane), sacral slope (SS; the angle between the sacral plate and the horizontal plane), and PI (the angle between the line perpendicular to the sacral plate and the line connecting the midpoint of the sacral plate to the bi-coxo-femoral axis). The lower-extremity parameters included in this analysis were the femur obliquity angle (FOA; the angle between the femoral shaft and the vertical line) [37] and knee flexion angle (KFA; the angle between the line from the hip axis to the midpoint of the bilateral notches of the femoral condyles and the line from the notch to the midpoint of the distal tibial joint surfaces) [38] (Figure 1).

The C2–C7 angle and the T1 slope range from 20–35° and 13–25° in healthy subjects, respectively [36,39]. The radiological criteria for ASD, according to the International Spine Study Group, include a frontal Cobb angle > 20°, SVA > 50 mm, TK > 60°, and/or PT > 25°. In healthy subjects, the KFA in standing position was <6° [38], and the knee joint flexion was defined as >6°.

### 2.3. Musculoskeletal Model

Full-body MSM was obtained using AMS (AMS. V. 6.0.5.4379; Anybody Technology, Alborg, Denmark). We previously developed an MSM in which the thoracic vertebrae were divided into 12 parts, and we modified muscle attachment points and pathways in the trunk and lower extremities based on magnetic resonance imaging data [27,28]. Furthermore, a cervical spine model, including seven vertebrae with three degrees of freedom spherical joints from T1 to C2, one degree of joint between C2 and the skull, and 136 muscles, was adopted [25] (Figure 2).

### 2.4. Spinopelvic and Lower-Extremity Sagittal Alignment Input and Simulation Process

The input for the entire spinal sagittal alignment was based on a previously reported method [29]. On lateral whole-spine radiographs, the center of the vertebral body was defined as the intersection of the diagonals of the quadrilateral formed by each vertebra (Figure 3).

The angle between the centers of each vertebra from C2 to S1 was then measured, and the respective values were input into the AMS. The pelvic and lower-extremity alignment inputs were similar to those in the previous method, and the posture of the model was corrected based on the patient-specific PT, SS, FOA, and KFA calculated on the radiographs [31]. This method was applied to all patients, and each patient’s body weight and height were input into the AMS (Figure 4).

### 2.5. Model Outputs

Using AMS, the intervertebral disc load at each vertebral level in the static standing posture was calculated using inverse dynamics analysis while considering known inertia and external forces. In the AMS, the disc load was calculated using the equation below:(1)$$F_{m}=f_{m}+\sum_{i=1}^{N}f_{i}$$

Here, $F_{m}$, $f_{m}$, *i*, *N*, and $f_{i}$ represented the compression force [*N*] at the intervertebral disc, the apparent force on the intervertebral joint, the muscle number, the total number of muscles, and the estimated muscle force of the simulation, respectively (Figure 5).

$F_{m}$ was estimated for each intervertebral segment from C2/C3 to L5/S1 and calculated by standardization according to each individual’s weight. The mean $F_{m}$ for each region was defined as follows: cervical $F_{m}$ from C2/C3 to C7/T1, upper thoracic $F_{m}$ from T1/2 to T3/4, lower thoracic $F_{m}$ from T4/5 to T12/L1, and lumbar $F_{m}$ from L1/2 to L5/S1.

Subjects were divided into two groups to compare the differences in $F_{m}$ at the cervical, upper thoracic, lower thoracic, and lumbar spine and each disc from C2/3 to L5/S1 between the low SVA (≤50 mm) and high SVA (>50 mm) groups (*n* = 7 per group). The cut-off values were based on the radiological criteria of the International Spine Study Group and Schwab’s realignment objectives for ASD [40].

### 2.6. Statistical Analysis

All continuous variables were expressed as mean ± standard deviation (SD). Comparisons between the two groups were performed using the Mann–Whitney U test for continuous variables. The relationship between $F_{m}$ and each parameter was examined using Spearman’s correlation coefficient (r). The results were verified using R version 3.6.2 (R Foundation for Statistical Computing, Vienna, Austria). Statistical significance was set at *p* < 0.05.

## 3. Results

Patient characteristics are shown in Table 1.

Compensatory changes, such as pelvic retroversion and knee flexion, were observed, and sagittal decompensation was observed as high COG–SVA and SVA (Table 1). The calculated $F_{m}$ (%BW) was 22.3 ± 7.3 at the cervical region, 30.8 ± 6.0 at the upper thoracic, 74.5 ± 11.8 at the lower thoracic, and 106.4 ± 44.5 at the lumbar.

A comparison of the two groups is presented in Table 2. The high SVA group had older age (*p* = 0.046), increased COG–SVA (*p* = 0.001), lower LL (*p* = 0.001), lower SS (*p* = 0.041) and increased SVA (*p* = 0.001). $F_{m}$ was significantly increased by 67.6% in the high SVA group (*p* = 0.046), but only in the lumber region (Table 2).

Figure 6 showed $F_{m}$ between the two groups at each disc level.

$F_{m}$ was higher in the high SVA group at the T4/5 (*p* = 0.008), T6/7 (*p* = 0.033), T12/L1 (*p* = 0.039), L1/2 (*p* = 0.044), and L5/S1 (*p* = 0.039) levels. The peak $F_{m}$ was in T12/L1 (88.4 ± 12.4) for the low SVA group and in L5/S1 (150.1 ± 55.9) for the high SVA group.

Table 3 showed the Spearman’s correlation coefficients between the mean $F_{m}$ at each region and spinopelvic sagittal alignment parameter.

There was a negative correlation between cervical $F_{m}$ with T1 slope (r = −0.589, *p* = 0.034) and lumbar $F_{m}$ with LL (r = −0.566, *p* = 0.035). Lumbar $F_{m}$ was positively correlated with COG–SVA (r = 0.615, *p* = 0.029), T1 slope (r = 0.613, *p* = 0.026), and SVA (r = 0.612, *p* = 0.020). There were no correlated parameters in the thoracic $F_{m}$.

Figure 7 showed two cases of low SVA and high SVA.

Figure 7A showed a lateral radiograph of an 80-year-old female with a C2–C7 angle of 7°, C2–C7 SVA of 19.1 mm, T1 slope of 12°, and SVA of 51 mm. Figure 7B showed the MSM input with the patient’s alignment parameters. Figure 7C showed an 86-year-old female with a C2–C7 angle of 33°, a C2–C7 SVA of 19.2 mm, a T1 slope of 41°, and an SVA of 139 mm. Figure 7D showed the MSM. For the first patient, cervical $F_{m}$ was 41.7 (%BW) and lumbar $F_{m}$ was 86.3 (%BW). For the second patient, cervical $F_{m}$ was 19.8 (%BW) and lumbar $F_{m}$ was 230.8 (%BW). In patients with a high T1 slope and high SVA, the cervical $F_{m}$ was lower but the lumbar $F_{m}\;$ was higher.

## 4. Discussion

In this study, the relationship between regional and global sagittal alignment parameters and disc loading was investigated using a patient-specific adapted MSM of the whole sagittal spinopelvic lower-extremity alignment for elderly women in standing posture. In the group of patients with SVA > 50 mm, the lumbar $F_{m}$ increased by 67.6%, and the maximum value of $F_{m}$ shifted to L5/S1. Cervical $F_{m}$ was negatively correlated with the T1 slope, lumbar $F_{m}$ was negatively correlated with LL, and SVA was positively correlated with the T1 slope. To the best of our knowledge, this is the first study to investigate the relationship between sagittal spinopelvic alignment and disc loading in patients with ASD. Our findings suggested that sagittal malalignment due to progressive spinal deformity resulted in further deformity due to increasing loading on thoracolumbar and lower lumbar discs; it was also associated with cervical disc loading.

Our main finding was the association between lumbar $F_{m}$ and sagittal alignment, with COG–SVA and SVA showing similar correlation coefficients for lumbar $F_{m}$. Clinically, it has been noted that there is an association between decreased LL with increased disc degeneration and increased lumbar degenerative disease with high SVA [41,42]. Measured intradiscal pressure increased with trunk forward bending and MSM analysis also reported decreased LL and trunk forward bending, which were associated with increased disc loading and vertebral compression loads in the thoracolumbar and lumbar regions [19,20,43,44,45,46,47,48]. Studies using AMS have examined the relationship between high SVA, lumbar kyphosis, and increased L4/5 and L5/S1 disc compression forces. The findings from our study were consistent with those from previous reports [30]. Increased lumbar disc load might be caused by increased erector spinae muscle activation due to inefficient standing posture by a forward shift of the gravity line and could lead to disc degeneration [30,41,48,49,50]. Furthermore, SVA has been used for global alignment because the C7 plumb line is considered a virtual COG. This parameter is easy to measure and is related to clinical outcomes [1,8,50]. However, recent studies revealed that COG is at the cranial center of mass in both normal subjects and patients with ASD [8,51]. Therefore, we evaluated the results using COG. In our study, SVA was correlated with lumbar $F_{m}$ to the same degree as COG–SVA, implying that SVA measurements were useful for investigating lumbar spinal disc loading. However, since lumbar $F_{m}$ has a large range, it is necessary to identify other factors that may affect it.

The negative correlation between T1 slope and cervical $F_{m}$ suggested that the T1 slope might be useful not only as an indicator of overall sagittal spinal balance, but also in elucidating the pathology of the cervical spine. The cervical spine is a complex region with the widest range of motion and is interrelated with the thoracic and lumbar vertebrae. Compensatory mechanisms act to keep the skull above the pelvis [52]. The T1 slope correlates with SVA and is useful in evaluating global sagittal balance. T1 slope has been reported to influence cervical lordosis [36,53,54]. The correlation between the C2–C7 angle and the T1 slope could be explained by the action of cervical spine extension with large cervical lordosis in ASD patients with decreased LL and high TK [51,53,55]. Although cervical kyphosis was reportedly associated with decreased health-related QoL, there are few studies on cervical spine disease in ASD patients, and no biomechanical studies have evaluated cervical disc loading and sagittal alignment. Therefore, the effect of increased lordosis remains unclear [13,53,54,56,57,58]. Conversely, a previous study analyzed the intervertebral disc load of all vertebrae, including the cervical spine, in physiological movements using AMS and reported that the cervical disc compression force increased in flexion and decreased in extension, while the disc compression force of the upper and middle thoracic spine showed only minor changes in flexion and extension [21]. As shown in Figure 7, patients with a high T1 slope extended the cervical spine and had a low cervical $F_{m}$, which could explain the negative correlation between T1 slope and cervical $F_{m}$. This result was consistent with a clinical study that indicated that the degree of cervical disc herniation and cervical spinal cord compression were negatively correlated with cervical lordosis and that the degree of disc herniation and disc height might recover as cervical lordosis does [59]. Our results implied that compensatory mechanisms for cervical extension in patients with ASD did not cause an increase in cervical disc. However, in this study, cervical $F_{m}$ did not correlate with the C2–C7 angle, nor did thoracic $F_{m}$ correlate with any alignment parameters, although disc compression force in T4/5 and T6/7 was significantly higher in the high SVA group. This result should take into consideration the clinical heterogeneity of ASD patients, who have various standing postures. In this regard, most of our participants had decreased LL, while few had severe cervical or thoracic deformities.

This study had several limitations. First, sample size was small (*n* = 14). Second, coronal alignment was not measured or adapted to the model, and the actual muscle activity and bone morphology of the individual was not considered in the analysis. Third, only a small number of patients had local deformities of the cervical and thoracic spine. Therefore, the analysis of the association between cervical and thoracic spine alignment and disc loading needs to be validated in other patient groups. However, in this study, we focused on determining the relationship between sagittal alignment and disc compression force in patients with ASD with various standing postures. The novelty of this study was that the analysis was performed considering COG and cervical alignment in ASD patients. Applying cervical spine alignment can improve the alignment of the model, as shown in Figure 4. In future large and age-matched samples, modified models’ analysis with individual muscle activity and bone morphology are needed. Furthermore, studies focusing on local deformities would be useful in revealing the relationship between spinal deformities and entire spinal disc loading.

## 5. Conclusions

In the group of patients with SVA > 50 mm, the lumbar disc compression force increased by 67.6%, and the maximum value of the disc compression force shifted to L5/S1. Cervical disc compression force was negatively correlated with T1 slope. Lumbar disc compression force was negatively correlated with LL and positively correlated with global alignment. Our findings suggested that sagittal malalignment due to spinal deformity resulted in further deformity due to increasing loading on thoracolumbar and lower lumbar discs; it was also associated with cervical disc loading. Future studies are needed to assess the relationship between regional deformities in the cervical thoracic spine and disc loading.

## Figures and Tables

**Figure 1 ijerph-19-16452-f001:**
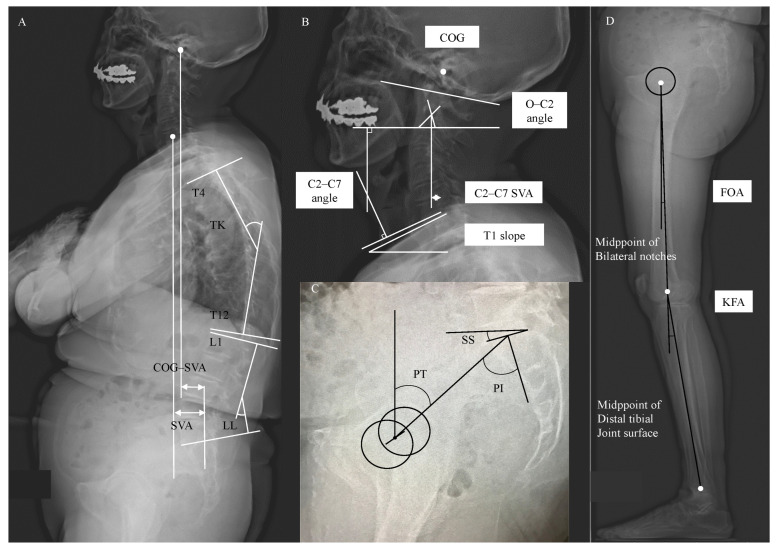
Sagittal alignment parameters in the standing position. (**A**,**B**) Spinal parameters, O–C2 angle (occipito–C2 angle), C2–C7 angle, C2–C7 sagittal vertical axis (C2–C7 SVA), the center of gravity of the head-posterior superior endplate of S1 (COG–SVA), T1 slope, thoracic kyphosis (TK), lumbar lordosis (LL), and SVA, were measured on a lateral whole-spine radiograph. (**C**) Pelvic parameters, PT (pelvic tilt), sacral slope (SS), and pelvic incidence (PI), were measured. (**D**) Lower-extremity parameters, the femur obliquity angle (FOA) and the knee flexion angle (KFA), were measured.

**Figure 2 ijerph-19-16452-f002:**
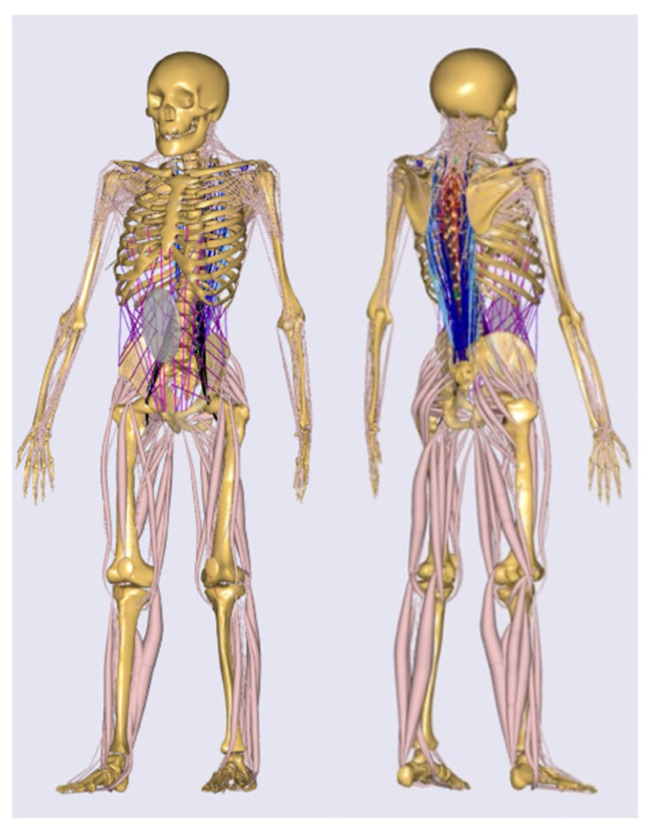
The constructed musculoskeletal model using the Anybody Modeling System. The constructed full-body model altered the muscle pathways and divided the cervical and thoracic spine.

**Figure 3 ijerph-19-16452-f003:**
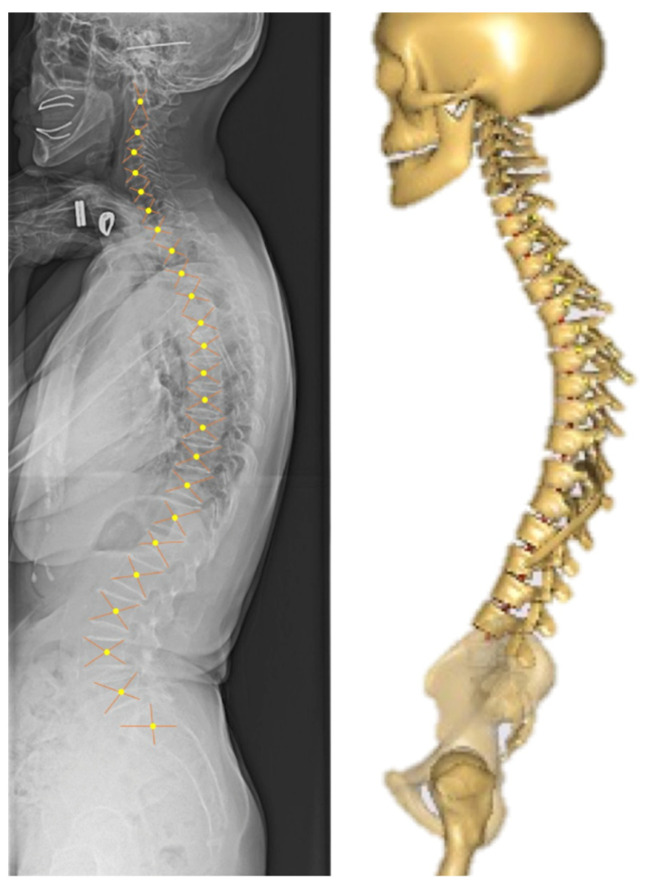
Defining the center of the vertebral body and modified musculoskeletal model. The musculoskeletal model was modified according to the sagittal spinal alignment of the patient.

**Figure 4 ijerph-19-16452-f004:**
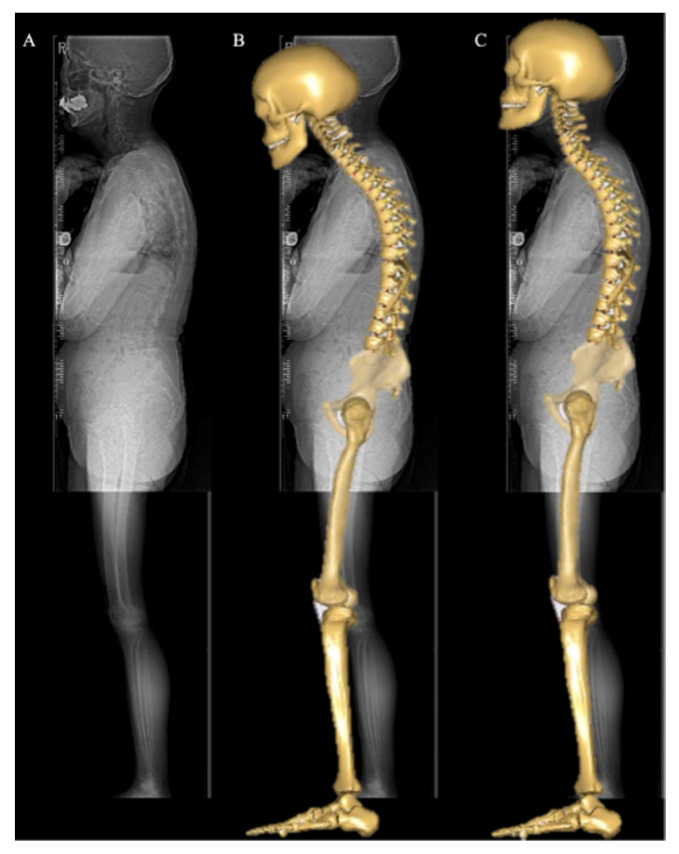
A patient’s full body lateral radiograph, and musculoskeletal model (MSM) images before and after cervical spine alignment correction. (**A**) Full-body lateral radiograph of a patient. (**B**) A modified MSM with altered thoracolumbar, pelvic, and lower-extremity alignment and unaltered cervical spine alignment. The model’s cranial location was not matched to the patient alignment. (**C**) In this study, a modified MSM with an altered whole spine, including cervical spine alignment, was used.

**Figure 5 ijerph-19-16452-f005:**
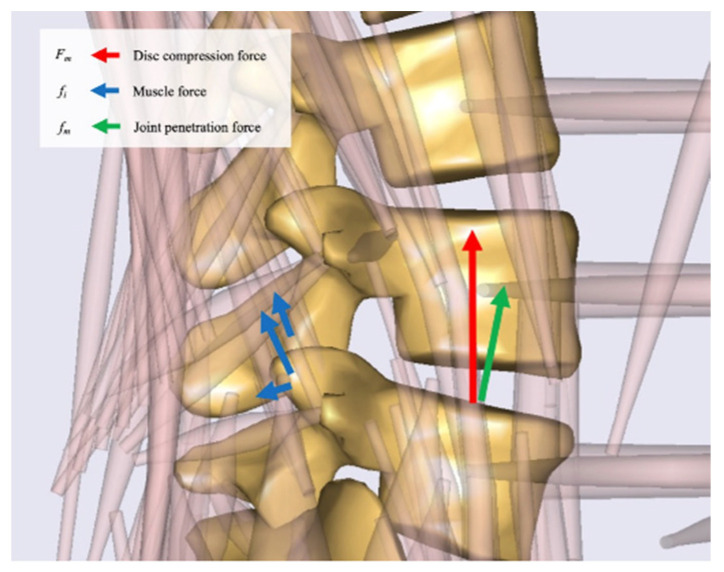
The magnified sagittal view of the lumbar spine in the musculoskeletal model. The green arrow indicates the penetration force exerted by an adjacent body segment. The blue arrow indicates the muscle forces acting on the vertebral body. The red arrow indicates the disc compression force, which was defined as the force perpendicular to the horizontal axis in the sagittal plane.

**Figure 6 ijerph-19-16452-f006:**
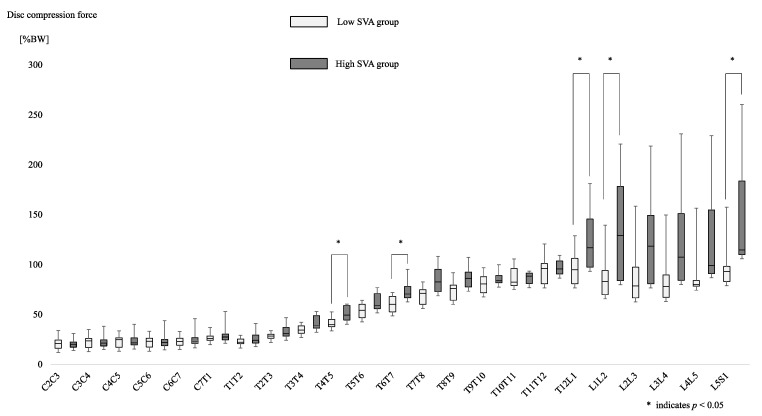
Compression force of the disc between the low SVA group and the high SVA group in each disc level. The light grey box plots indicate the disc compression force in each intervertebral body in the low SVA group, and the dark grey box plots indicate this for the high SVA group.

**Figure 7 ijerph-19-16452-f007:**
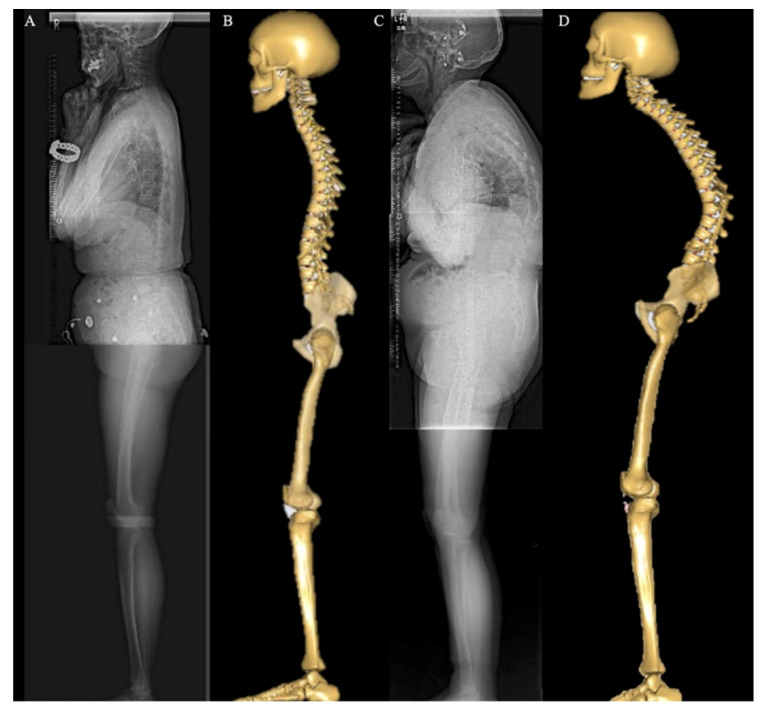
Example cases of low SVA and high SVA patients. (**A**) Lateral radiographs of the whole spine and lower extremity in the standing position in an 80-year-old female. (**B**) The musculoskeletal model with input of the patient’s alignment parameters. (**C**) Lateral radiographs of an 86-year-old female. (**D**) The musculoskeletal model of the patient.

**Table 1 ijerph-19-16452-t001:** Clinical information of all patients.

Variable	Value
Total patients (*n*)	14
Age (years)	78.8 ± 7.0 (68–92)
BMI (kg/m^2^)	22.9 ± 4.3 (16.4–32.8)
O–C2 angle (°)	12.3 ± 7.6 (0–28)
C2–C7 angle (°)	21.2 ± 11.0 (6–42)
C2–C7 SVA (mm)	21.0 ± 8.0 (7.6–38.7)
COG–SVA (mm)	86.3 ± 55.6 (15–224.7)
T1 slope (°)	29.4 ± 11.6 (12–58)
TK (°)	36.4 ± 15.2 (12–67)
LL (°)	38.2 ± 15.8 (15–69)
PT (°)	24.5 ± 8.7 (12–47)
SS (°)	27.8 ± 6.7 (13–38)
PI (°)	52.9 ± 9.5 (30–65)
PI−LL (°)	10.9 ± 19.5 (−23–41)
SVA (mm)	66.1 ± 52.5 (4–162)
FOA (°)	5.8 ± 3.0 (0–13)
KFA (°)	8.5 ± 4.5 (0.3–18)

Values were expressed as the mean ± standard deviation (range). BMI, body mass index; O–C2 angle, occipito–C2 angle; COG, center of gravity; SVA, sagittal vertical axis; TK, thoracic kyphosis; LL, lumbar lordosis; PT, pelvic tilt; SS, sacral slope; PI, pelvic incidence; FOA, femur obliquity angle; KFA, knee flexion angle.

**Table 2 ijerph-19-16452-t002:** Comparison of the low SVA and high SVA groups.

Variables	Low SVA Group (*n* = 7)	High SVA Group (*n* = 7)	*p*-Value
Age (years)	74.0 ± 6.7	82.1 ± 6.0	0.046 *
BMI (kg/m^2^)	25.4 ± 3.5	22.0 ± 3.8	0.142
O–C2 angle (°)	12.7 ± 7.9	12.8 ± 7.3	0.980
C2–C7 angle (°)	18.4 ± 12.3	21.3 ± 10.7	0.685
C2–C7 SVA (mm)	17.1 ± 8.1	24.7 ± 7.0	0.089
COG–SVA (mm)	45.5 ± 15.5	134.0 ± 47.4	<0.001 *
T1 slope (°)	24.2 ± 5.7	35.5 ± 13.7	0.070
TK (°)	33.7 ± 8.4	35.7 ± 19.0	0.820
LL (°)	50.6 ± 11.7	27.9 ± 12.5	<0.001 *
PT (°)	23.6 ± 7.0	24.7 ± 9.7	0.819
SS (°)	32.2 ± 4.5	24.6 ± 6.8	0.041 *
PI (°)	55.9 ± 7.6	50.0 ± 10.9	0.338
PI−LL (°)	5.3 ± 13.0	15.0 ± 23.4	0.396
SVA (mm)	21.1 ± 13.7	109.9 ± 39.5	0.001 *
FOA (°)	4.3 ± 3.1	6.7 ± 2.5	0.166
KFA (°)	7.4 ± 4.6	9.4 ± 4.6	0.463
Cervical $F_{m}$ (%BW)	23.9 ± 6.5	24.3 ± 7.9	0.912
Upper thoracic $F_{m}$ (%BW)	29.3 ± 4.5	33.6 ± 6.9	0.225
Lower thoracic $F_{m}$ (%BW)	70.7 ± 10.4	82.5 ± 9.3	0.067
Lumbar $F_{m}$ (%BW)	79.2 ± 8.9	132.8 ± 52.3	0.046 *

Values were expressed as the mean ± standard deviation; * indicates significant findings. BMI, body mass index; O–C2 angle, occipito–C2 angle; COG, center of gravity; SVA, sagittal vertical axis; TK, thoracic kyphosis; LL, lumbar lordosis; PT, pelvic tilt; SS, sacral slope; PI, pelvic incidence; FOA, femur obliquity angle; KFA, knee flexion angle; $F_{m}$, disc compression force.

**Table 3 ijerph-19-16452-t003:** Spearman’s correlation coefficients between the mean disc compression force at each part and spinopelvic sagittal alignment parameter in all patients.

Variables	$\mathrm{Cervical}\;F_{m}$		$\mathrm{Upper}\;\mathrm{Thoracic}\;F_{m}$		$\mathrm{Lower}\;\mathrm{Thoracic}\;F_{m}$		$\mathrm{Lumbar}\;F_{m}$	
	r	*p*-Value	r	*p*-Value	r	*p*-Value	r	*p*-Value
O–C2 angle	−0.152	n.s.	0.089	n.s.	−0.149	n.s.	0.116	n.s.
C2–C7 angle	−0.456	n.s.	−0.203	n.s.	−0.022	n.s.	0.247	n.s.
C2–C7 SVA	−0.108	n.s.	0.305	n.s.	0.165	n.s.	0.451	n.s.
COG–SVA	−0.121	n.s.	0.385	n.s.	0.280	n.s.	0.615	0.029 *
T1 slope	−0.589	0.034 *	0.028	n.s.	0.245	n.s.	0.613	0.026 *
TK	−0.203	n.s.	0.090	n.s.	0.269	n.s.	0.176	n.s.
LL	0.007	n.s.	−0.363	n.s.	−0.416	n.s.	−0.566	0.035 *
PT	−0.168	n.s.	0.289	n.s.	0.187	n.s.	0.119	n.s.
SS	0.304	n.s.	0.255	n.s.	0.092	n.s.	−0.139	n.s.
PI	−0.033	n.s.	0.426	n.s.	0.072	n.s.	0.028	n.s.
PI−LL	−0.165	n.s.	0.455	n.s.	0.455	n.s.	0.332	n.s.
SVA	−0.180	n.s.	0.317	n.s.	0.310	n.s.	0.612	0.020 *

* Indicates significant findings; n.s. indicates not significant.
$\;F_{m}$, disc compression force; r, Spearman’s correlation coefficient; O–C2 angle, occipito–C2 angle; COG, center of gravity; SVA, sagittal vertical axis; TK, thoracic kyphosis; LL, lumbar lordosis; PT, pelvic tilt; SS, sacral slope; PI, pelvic incidence.

## Data Availability

Not applicable.

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
