# Peer review of "Relationship between Intervertebral Disc Compression Force and Sagittal Spinopelvic Lower Limb Alignment in Elderly Women in Standing Position with Patient-Specific Whole Body Musculoskeletal Model"

_ijerph, 2022, doi:10.3390/ijerph192416452_

Round 1

Reviewer 1 Report

This study investigated the relationship between sagittal alignment parameters and disc compression force in elderly women using a patient-specific whole musculoskeletal model. The manuscript is well written and the findings are important and interesting to the field. Here are my comments/questions: 

1.    Line 186: Please add the information about which table/figure the reader can find the results.

2.    Figure 6: Please explain why the compression force has wide range in each lumbar level, especially in high SVA group. The authors also observe significant changes of compression force in specific disc levels between low and high SVA groups. Does the difference in disc levels where the largest curvature forms affect the compression force?  

3.    Excessive loading predisposes intervertebral disc to degenerative changes. What do the authors think of the relationship between compression force and intervertebral disc degeneration in SVA group?  

4.    Table 3: “Title 3”, “FOA, femur obliquity angle”, and “KFA, knee flexion angle” should be deleted.

5.    Table 3: Please clarify if the correlation coefficients between Fm and sagittal alignment parameter are done in all the samples or just high SVA group. 

6.    The compression force is higher in T4T5 and T6T7 levels in high SVA group compared to low SVA, but no correlation is found between thoracic Fm and any of the sagittal alignment parameter.  

7.    Line 240 and 310: The sentence does not make sense. It should be “due to increasing loading on…”.

Author Response

December 3rd, 2022

We thank you and the reviewers for your thoughtful suggestions and insights, which have substantially enriched the manuscript and produced a better account of the research. We hope that the revised manuscript is now suitable for publication in your journal.

The manuscript has been carefully rechecked and appropriate changes have been made in accordance with the reviewers’ suggestions. The responses to the comments have been prepared and attached herewith. The revisions are indicated in red font in the revised manuscript. Furthermore, the revised manuscript has been checked by a native English speaker.

Reviewer 1

This study investigated the relationship between sagittal alignment parameters and disc compression force in elderly women using a patient-specific whole musculoskeletal model. The manuscript is well written and the findings are important and interesting to the field. Here are my comments/questions: 

  1. Line 186: Please add the information about which table/figure the reader can find the results.

Response: We thank the reviewer for this valuable suggestion. We have added this information.

  1. Figure 6: Please explain why the compression force has wide range in each lumbar level, especially in high SVA group. The authors also observe significant changes of compression force in specific disc levels between low and high SVA groups. Does the difference in disc levels where the largest curvature forms affect the compression force?  

Response: We thank the reviewer for this important suggestion.

Since there is a correlation between lumbar lordosis and disc compression force, the largest curvature may affect compression force. Indeed, Bassani et al. reported that disc compression force changed at the L4/5 and L5/S1 levels by altering the sacral slope1). However, since this study included patients with varying alignments, we cannot conclude about the relationship between some geometries and disc compression force. Therefore, in the discussion we added the following sentence:

However, since lumbar  has a large range, it is necessary to identify other factors that may affect it.” (Page10, line 256–257)

1)Bassani T, Casaroli G, Galbusera F. Dependence of lumbar loads on spinopelvic sagittal alignment: An evaluation based on musculoskeletal modeling. PLoS One. 2019 Mar 18;14(3):e0207997. doi: 10.1371/journal.pone.0207997. PMID: 30883563; PMCID: PMC6422292.

  1. Excessive loading predisposes intervertebral disc to degenerative changes. What do the authors think of the relationship between compression force and intervertebral disc degeneration in SVA group? 

Response: We thank the reviewer for this valuable comment. 

We think “Our findings suggested that sagittal malalignment due to progressive spinal deformity resulted in further deformity due to increasing loading on thoracolumbar and lower lumbar discs; it was also associated with cervical disc loading.” However, we did not highlight the relationship between high SVA and disc degeneration. Therefore, we add the following sentence: “Increased lumbar disc load might be caused by increased erector spinae muscle activation due to inefficient standing posture by a forward shift of the gravity line and could lead to disc degeneration.” (Page10, line 249–250)

  1. Table 3: “Title 3”, “FOA, femur obliquity angle”, and “KFA, knee flexion angle” should be deleted.

Response: We thank the reviewer for this valuable comment. We deleted the words from Table 3.

  1. Table 3: Please clarify if the correlation coefficients between Fm and sagittal alignment parameter are done in all the samples or just high SVA group. 

Response: We thank the reviewer for this very pertinent suggestion. The analysis was performed in all patients, and we have changed the title to clarify this. (Page, line 204)

  1. The compression force is higher in T4T5 and T6T7 levels in high SVA group compared to low SVA, but no correlation is found between thoracic Fm and any of the sagittal alignment parameter.  

Response: We thank the reviewer for this valuable comment.

We highlight the difference between two groups and correlation coefficient analysis in the Discussion section as follows:

“However, in this study, cervical  did not correlate with the C2-C7 angle, nor did thoracic  correlate with any alignment parameters, although disc compression force in T4/5 and T6/7 was significantly higher in the high SVA group.” (Page10–11, line 283–287)

  1. Line 240 and 310: The sentence does not make sense. It should be “due to increasing loading on…”.

Response: We thank the reviewer for this valuable suggestion. We changed the sentences as you suggested.

Thank you for your consideration. We look forward to hearing from you.

Reviewer 2 Report

Authors have analyzed the relationship between sagittal alignment and disc compression force (Fm) in fourteen elderly females with a known history of osteoporosis. The authors showed the relationship between regional and global sagittal alignment parameters and also investigated disc loading through the musculoskeletal model (MSM). Overall this is an interesting study, and the findings are new. However, there are a few limitations of the study  

o   less number of subjects, and all are elderly

o   Most of the findings are based on radiographs; muscle activity and bone morphology of the individual were not taken into consideration 

o   Age-matched control subjects were not radiographed 

o   Aging seems to be one of the reasons for intervertebral disc compression and could affect the study’s overall conclusion. 

Author Response

December 3rd, 2022

We thank you and the reviewers for your thoughtful suggestions and insights, which have substantially enriched the manuscript and produced a better account of the research. We hope that the revised manuscript is now suitable for publication in your journal.

The manuscript has been carefully rechecked and appropriate changes have been made in accordance with the reviewers’ suggestions. The responses to the comments have been prepared and attached herewith. The revisions are indicated in red font in the revised manuscript. Furthermore, the revised manuscript has been checked by a native English speaker.

Reviewer 2

Authors have analyzed the relationship between sagittal alignment and disc compression force (Fm) in fourteen elderly females with a known history of osteoporosis. The authors showed the relationship between regional and global sagittal alignment parameters and also investigated disc loading through the musculoskeletal model (MSM). Overall this is an interesting study, and the findings are new. However, there are a few limitations of the study  

  1. less number of subjects, and all are elderly

Response: We thank the reviewer for this valuable comment. As you pointed out, a smaller number of subjects and all elderly patients are limitations in this study. However, patient age does not affect the analysis of musculoskeletal model. Moreover, the measured studies of intervertebral discs pressure have been performed on healthy subjects, and there are no studies on the elderly or age-related changes in spinal deformity. The pathophysiology and biomechanical analysis of lower back pain and degenerative progression in these patients are not fully understood, and we believe that the findings of this study will be useful to this end.

  1. Most of the findings are based on radiographs; muscle activity and bone morphology of the individual were not taken into consideration 

Response: We thank the reviewer for this valuable comment. In order to perform highly accurate analysis of individual patients, it is necessary to construct a model that considers muscle activity and bone morphology. However, since the purpose of this study was to investigate changes in sagittal alignment and intervertebral disc compression force, we did not change the bone shape. It is possible that muscle activity also changes due to changes in alignment, but since muscle activity has not been analyzed, it has been added to the limitations as a future research topic as follows:

In future large and age-matched samples, modified models’ analysis with individual muscle activity and bone morphology are needed. Furthermore, studies focusing on local deformities would be useful in revealing the relationship between spinal deformities and entire spinal disc loading.” (Page 11, line 300–301)

  1. Age-matched control subjects were not radiographed 

Response: We thank the reviewer for this insightful comment.

Absences of age-matched control subjects is one of the limitations of this study.

However, as we mentioned in comment #1, aging does not affect the musculoskeletal model analysis. Furthermore, including the elderly was necessary to analyze the effects of alignment changes due to aging. Since muscle atrophy and changes in bone morphology due to aging are not taken into consideration, future research is necessary as described in comment #2.

  1. Aging seems to be one of the reasons for intervertebral disc compression and could affect the study’s overall conclusion.

Response: We thank the reviewer for this important comment. As you pointed out, aging leads to degeneration of intervertebral discs, for which we think age-matching is valuable. However, as we mentioned in comments #2 and #3, aging does not affect the musculoskeletal model analysis. Clinically, it is necessary to consider the effect of back muscle atrophy due to aging, and in the future, it is necessary to adapt muscle activity as described in the limitations.

Thank you for your consideration. We look forward to hearing from you.

Reviewer 3 Report

Although you have commented on the limitations, the sample is insufficient, there is not enough sample to be able to polarise the results to the population. Too few subjects in each group. How do you justify this?

In the conclusion, delete the first sentence. No need to repeat what you have done.

Author Response

December 3rd, 2022

We thank you and the reviewers for your thoughtful suggestions and insights, which have substantially enriched the manuscript and produced a better account of the research. We hope that the revised manuscript is now suitable for publication in your journal.

The manuscript has been carefully rechecked and appropriate changes have been made in accordance with the reviewers’ suggestions. The responses to the comments have been prepared and attached herewith. The revisions are indicated in red font in the revised manuscript. Furthermore, the revised manuscript has been checked by a native English speaker.

Reviewer 3

Although you have commented on the limitations, the sample is insufficient, there is not enough sample to be able to polarise the results to the population. Too few subjects in each group. How do you justify this?

In the conclusion, delete the first sentence. No need to repeat what you have done.

Response: We thank the reviewer for this valuable suggestion. As you pointed out, the small sample size is one of the limitations of this study.

However, model studies can be analyzed regardless of individual patients’ age, weight, muscles, and bone morphology. Therefore, unlike clinical studies, it is possible to calculate data without being affected by factors other than alignment, for which we believe that a high power can be obtained, even with a small sample size.

In fact, even in other musculoskeletal model studies, the number of subjects is 12–141-2). Therefore, we believe that the number of patients in this study is justified. We added the following sentences in the limitations:

In future large and age-matched samples, modified models’ analysis with individual muscle activity and bone morphology are needed.” (Page 11, line 300–301)

1)Leijendekkers RA, Marra MA, Kolk S, et al. Gait symmetry and hip strength in women with developmental dysplasia following hip arthroplasty compared to healthy subjects: A cross-sectional study. PLoS One. 2018;13(2):e0193487. Published 2018 Feb 23. doi:10.1371/journal.pone.0193487

2) Theodorakos I, Healy A, Chatzistergos P, Andersen MS, Chockalingam N. Assessment of the effect of a total contact cast on lower limb kinematics and joint loading. Gait Posture. 2022;98:203-209. doi:10.1016/j.gaitpost.2022.09.075

Thank you for your consideration. We look forward to hearing from you.

Round 2

Reviewer 2 Report

The authors have responded to all queries and the manuscript can be accepted.